# OpenReview forum: "On the Global Optimality of Policy Gradient Methods in General Utility Reinforcement Learning"
_NeurIPS.cc/2025/Conference — NeurIPS 2025 poster_

### Official Review · Reviewer_Fepe · 2025-06-18

**Clarity:** 3
**Significance:** 3
**Originality:** 3
**Rating:** 5
**Confidence:** 4

**Summary:**

The paper establishes the global convergence of policy gradient in RL with concave utility functions. Further it provides sample complexity for the same, approximating the occupany measures.

**Questions:**

**Q1). Global convergence requires the mis-matchcoefficient to be fininte  which is typically satisfied by taking assuming initial state distribution $\min_{s}\mu(s)>0$ (Xiao 2022). Otherwise, it has been shown that policy gradient can (in linear RL) converges to some local soution [1]. The paper claims their result to be applicable to the continuous/infinite state space, it is not clear how this mismatch coefficient can be finite in these cases.**


[1] @misc{koren2025policygradienttreesearch,
      title={Policy Gradient with Tree Search: Avoiding Local Optimas through Lookahead},
      author={Uri Koren and Navdeep Kumar and Uri Gadot and Giorgia Ramponi and Kfir Yehuda Levy and Shie Mannor},
      year={2025},
      eprint={2506.07054},
      archivePrefix={arXiv},
      primaryClass={cs.LG},
      url={https://arxiv.org/abs/2506.07054},}

**Q2). Proposition 1 of the paper establishes sample complexity of $O(\epsilon^{-2})$ for estimating occupation measure. I wonder, Lemma 1 of Kumar et al. 2022 that shows $\gamma$-contracting operator for computing occupation measure (bootstraping just like Bellman operator), if this operator can help increase the sample complexity of the occupation measure?**

**Q3 Could this work be extended to average reward case as studied in [4]? What could be possible challenges?**

[4] @inproceedings{
kumar2025global,
title={Global Convergence of Policy Gradient in Average Reward {MDP}s},
author={Navdeep Kumar and Yashaswini Murthy and Itai Shufaro and Kfir Yehuda Levy and R. Srikant and Shie Mannor},
booktitle={The Thirteenth International Conference on Learning Representations},
year={2025},
url={https://openreview.net/forum?id=2PRpcmJecX}
}

**Q4). (Bonus question) What do authors think about the sample complexity of single-time actor critic RELU as studied in [2,3]?**

[2]@misc{chen2024finitetimeanalysissingletimescaleactorcritic,
      title={Finite-time analysis of single-timescale actor-critic},
      author={Xuyang Chen and Lin Zhao},
      year={2024},
      eprint={2210.09921},
      archivePrefix={arXiv},
      primaryClass={cs.LG},
      url={https://arxiv.org/abs/2210.09921},
}

[3]@misc{kumar2025convergencesingletimescaleactorcritic,
      title={On the Convergence of Single-Timescale Actor-Critic},
      author={Navdeep Kumar and Priyank Agrawal and Giorgia Ramponi and Kfir Yehuda Levy and Shie Mannor},
      year={2025},
      eprint={2410.08868},
      archivePrefix={arXiv},
      primaryClass={cs.LG},
      url={https://arxiv.org/abs/2410.08868},
}

**Ethical Concerns:**

["NO or VERY MINOR ethics concerns only"]

**Final Justification:**

The paper has nice contribution to convex RL, hence worth accepting.

**Limitations:**

Yes.

**Paper Formatting Concerns:**

There are few minor typos and the presentation of the paper could be improved. But otherwise everything seems ok.

**Quality:**

3

**Strengths And Weaknesses:**

Strength: Theorem 1 is the most significant result of the paper. It establishes gradient domination for RLGU. Similar to linear RL, it upper bounds  the sub-optimality (a global quantity) with current  gradient (a local quantity).   Combined with sufficient increase lemma (smooth function), this result directly implies global convergece of policy gradient of RLGU with an iteration complexity of $O(\frac{1}{\epsilon})$ similar to linear RL. The proof for the theorem is also very short and sweet.  In my opinion, this result on its own is enough for the paper to be accepted.

 Weakness: (Minor) The paper misses to cite [1,2] that establishes convergence of policy gradient in finite steps (independent of state space).

[1]@misc{liu2024elementaryanalysispolicygradient,
      title={Elementary Analysis of Policy Gradient Methods},
      author={Jiacai Liu and Wenye Li and Ke Wei},
      year={2024},
      eprint={2404.03372},
      archivePrefix={arXiv},
      primaryClass={math.OC},
      url={https://arxiv.org/abs/2404.03372},
}

[2] @misc{liu2024convergenceprojectedpolicygradient,
      title={On the Convergence of Projected Policy Gradient for Any Constant Step Sizes},
      author={Jiacai Liu and Wenye Li and Dachao Lin and Ke Wei and Zhihua Zhang},
      year={2024},
      eprint={2311.01104},
      archivePrefix={arXiv},
      primaryClass={math.OC},
      url={https://arxiv.org/abs/2311.01104},
}

---

> ### Author Rebuttal · Authors · 2025-07-31
>
> We thank the reviewer for their time and for their thoughtful comments. We appreciate their positive feedback. We also agree with the reviewer that the gradient domination result is probably our most important contribution. We answer their questions below.
>
> > **Weakness 1 (minor):** The paper misses to cite [1,2] that establishes convergence of policy gradient in finite steps (independent of state space).
>
>
> **Response to Weakness 1 (minor):**  We thank the reviewer for mentioning these interesting references, we will definitely add them to our related work discussion about the convergence of PG methods for classical RL.
>
> > **Question 1:** Global convergence requires the mismatch coefficient to be finite, which is typically satisfied by choosing the initial state distribution to cover the optimal policy’s support (as in Xiao 2022). For continuous/infinite state spaces, how can this coefficient remain finite?
>
> **Response to Question 1 (continuous setting):** We assume throughout the paper that the state and action spaces are finite as we mention in l.100. In particular Theorem 1 in section 3 holds under this setting and we also make the assumption that $\mu$ has full support in theorem 1 as mentioned by the reviewer (see the theorem statement). While we mention the applicability of **our algorithm** to continuous state spaces in practice (in remark 2, p.6, section 4), the current analysis does not extend to that setting as noticed by the reviewer. We will make this clearer and we thank the reviewer for the recent reference which we will add.
>
> > **Question 2:** Proposition 1 gives sample complexity for estimating the occupancy measure. Could sample complexity be improved using a contracting operator, as in Kumar et al. 2022?
>
> **Response to Question 2:** This is an interesting question and idea. We have briefly thought about this. However, the Bellman consistency equation satisfied by the occupancy measure is unfortunately backward and not forward like in standard Bellman operators which allow for forward sampling. We briefly discuss this issue in l. 193-197 in the main part and expand on it in l. 659-668 in the appendix p. 17 (see there for the exact recursion satisfied by the occupancy measure which makes it clear than one cannot use forward sampling). Therefore even if it is a contraction, forward sampling using that Bellman consistency equation is not possible. As a consequence, we believe that Algorithm 1 (as proposed in the reference provided by the reviewer for the occupancy measure estimation step) cannot be implemented with standard bootstrapping. The reason is that we believe there is a missing transposition of the transition matrix $P^{\pi}$ which should be  $P^{\pi,T}$ in Lemma 1. There might be some other way to exploit the recursion, we have mentioned the work of Hallak, A. and Mannor, S. (2017). 'Consistent on-line off-policy evaluation' which might be relevant in this regard. This requires further investigation.
>
> > **Question 3:** Could your analysis extend to the average‑reward setting (e.g., Kumar et al. 2025)? What are the challenges?
>
> **Response to Question 3:** This is definitely an interesting question that is worth investigating in future work, especially that the analysis of PG algorithms is quite recent for the average reward setting as mentioned by the reviewer. We will add this reference to our related work discussion. We foresee that this extension is possible, one needs to be careful with the new policy gradient for that setting using differential value functions, use the appropriate smoothness result recently developed and derive adequate gradient domination results.
>
> > **Question 4 (bonus):** What do you think about the sample complexity of single‑timescale actor–critic methods, as studied by Chen & Zhao 2024 and Kumar et al. 2025?
>
> **Response to Question 4:**   We have restricted our literature review discussion to works either addressing RLGU or PG methods (without critics, i.e. excluding AC methods). We will definitely add these two references for completeness. In our RLGU setting, we do not deal with AC methods even if occupancy measure estimation can be seemingly seen as a critic step. Recall that a Bellman like update for occupancy measure estimation is not available to us as we discussed above and therefore ideas from the interesting single time scale analysis you mentioned cannot be immediately used. On a side note, the question of the sample complexity for global optimality for single time scale AC is interesting and the paper you mentioned seems to leave open an interesting gap with the $O(\epsilon^{-2})$ sample complexity.
>
>
> We appreciate the reviewer’s detailed and supportive assessment. We will incorporate the suggested citations and clarifications in the final version.

---

> > ### Comment · Reviewer_Fepe · 2025-08-01
> > **Reviewer response to author rebuttal.**
> >
> > I thank authors for the rebuttal.  I maintain positive attitude towards the paper.

---

> > > ### Author Response · Authors · 2025-08-06
> > > **Thank you**
> > >
> > > We sincerely appreciate your time and valuable feedback. Thank you for your continued support and positive assessment of our work.

---

### Official Review · Reviewer_hs2m · 2025-06-29

**Clarity:** 3
**Significance:** 3
**Originality:** 3
**Rating:** 5
**Confidence:** 3

**Summary:**

This paper addresses reinforcement learning with general utilities, in which the objective is a (potentially non-linear) function of the state-action occupancy measure. The paper provides a global optimality result for policy gradient when the utility is concave and the MDP is tabular through gradient domination technique. Next, the paper addresses the same problem beyond the tabular setting, where the state-action occupancy measure is estimated via maximum likelihood within an approximating class. In this more challenging settings, first-order and global optimality guarantees, together with corresponding sample complexities, are derived under some more restrictive assumptions.

**Questions:**

**Tabular setting.**

(1) I have some confusion about the strength of gradient domination result. From my understanding, Theorem 1 is saying that gradient domination for every pseudoreward gives a global optimality result. This makes a lot of sense, as Hazan et al. have established that RLGU can be cast as a sequence of (linear) RL problems, for which gradient domination leads to global optimality of PG as know from previous works. However, the question here is how much the distribution shift constant is bigger here w.r.t. standard gradient domination. From what I know, the constant is usually thought with $\mu$ as the state distribution of the initial policy parametrization, which needs to cover an optimal policy state distribution. Here, we need a distribution $\mu$ that covers all of the policies that are optimal for some pseudoreward (?) In general, this looks like the set of all the deterministic policies, which would make the constant very large and the result somewhat vacuous. Am I missing something? I would recommend the authors to make an explicit comparison between the gradient domination analysis in RL (e.g., Lemma 4.1 in Agarwal et al., 2021) and RLGU in the manuscript.

**Large-scale setting.**

(2) Assumptions. Understanding the assumptions here is crucial to assess the generality of the results and when do the guarantees hold. However, the discussion of the assumptions is somewhat lacking in my opinion. Realizability (Assumption 2) is ubiquitous in RL theory for value function estimation. I am wondering how stronger it can be for occupancy estimation. Assumption 4 also could be explained beyond saying "This assumption captures most of the problems of interest in RLGU and beyond". Why is that the case? Assumption 5 is quite hard to process, I don't think I understood the implications of any of these. Finally, the $\epsilon_{MLE}$ assumption introduced in the statement of Theorem 3 warrant some discussion. Isn't this implying that the algorithm doesn't really converge ($\epsilon_{MLE}$ is just a constant error that is not scaled by N)? Moreover, why $\epsilon_{MLE}$ doesn't appear in Corollary 2? Shall not that be something like "for any precision $\epsilon > \epsilon_{MLE}$..."?

(3) Occupancy measure estimation. Assuming i.i.d. sampling is somewhat problematic. It means $n$ is actually the number of trajectory, not state-action pairs. The authors may consider making this more explicit in the paper, adding proper factors to the sample complexity results (at least an $H$ factor shall be needed). Perhaps the results could look be refined by using more refined analysis for  sampling and estimation from Markov chains. For the undiscounted case "Hoeffding's inequality for general Markov chains and its applications to statistical learning" Fan et al. 2021. For the discounted case "A tale of sampling and estimation in discounted reinforcement learning" Metelli et al. 2023.

(4) I am not very familiar with the analysis of Barakat et al. 2023. The paper here says that this previous work doesn't provide global optimality guarantees but just first-order. However, I am wondering whether their analysis would translate to global optimality when the same assumptions of this paper are considered. Aside from MSE and MLE step, what makes Barakat et al. analysis break for global optimality?

**Other minor questions/comments.**

(5) The results in the paper are provided for the discounted setting. In principle, can we use the same technical tools to study undiscounted settings, with finite or infinite horizon?

(6) "Assumption 2 is satisfied for instance for the class of generalized linear models". Well, this is not supported when realizability is included in the assumption. Better to clarify in the text.

(7) The discussion of the related work looks mostly complete. Perhaps "Regret minimization for reinforcement learning with vectorial feedback and complex objectives" Cheung 2019 and the pair "Submodular reinforcement learning" Prajapat et al. 2023, "Global reinforcement learning: Beyond linear and convex rewards via submodular semi-gradient methods" De Santi et al. 2024 could also be mentioned.

**Ethical Concerns:**

["NO or VERY MINOR ethics concerns only"]

**Final Justification:**

I was missing some information on the novelty of the analysis and the needed assumptions, which have been covered by the authors' response. I am now more convinced on the value of the paper, thus increasing my score from borderline accept (4) to accept (5)

**Limitations:**

On the limitations, see questions above. Potential negative societal impact are not applicable at this level of abstraction.

**Paper Formatting Concerns:**

/

**Quality:**

4

**Strengths And Weaknesses:**

**Strengths.**
- RLGU is a framework that unifies a large body of settings of interest for RL, so any advancement on its understanding is potentially valuable;
- The global optimality results for PG in tabular and larger scale RLGU settings are novel to the best of my knowledge;
- The paper makes good use of recent advancements in the analysis policy gradient for RL to establish more general results beyond linear utilities;
- The paper is well-written, with a mostly solid presentation.

**Weaknesses.**
- For both the tabular and larger scale settings, the significance of the results could be discussed in more details. It is not fully clear to which extent the reported results are a direct implication of previous results in RLGU and analysis of PG. While there might be important value in stating those results explicitly, still showing that the combination of what is previously known is non-trivial would be desirable;
- The tabular setting could enjoy a more direct comparison with the analysis of the linear utility case;
- The results with occupancy measure approximation is built upon a long string of assumptions. Some are more reasonable than others, but it would help to give more intuition/explanations on when they hold, why they are important, and how they might be overcome;
- Some slippery details are briefly mentioned in the text and then buried in the appendix, most of all the need for i.i.d. state samples for MLE of the occupancy measure.

**Evaluation.**
Overall this looks like a solid and technically sound paper on a very interesting topic. While the readers who are familiar with previous results in RLGU may find some of the results not particularly surprising, seeing them stated explicitly and neatly in the theorem statements is great. However, as it is often the case in these kind of analysis, I think the assumptions are the core of the results, and I would like to see more discussion on them before judging the extent to which the paper is advancing the literature here. I am currently providing a borderline score, but I am open to increase if the authors can reassure me on the generality and strength of their results.

---

> ### Author Rebuttal · Authors · 2025-07-31
>
> We thank the reviewer for their time, their thorough and detailed review, and their thoughtful comments. We are glad the reviewer found our paper 'well-written, with a mostly solid presentation' and 'a solid and technically sound paper on a very interesting topic'. We respond to each one of the reviewer's concerns and questions in the following.
>
> >**Question 1:** I have some confusion about the strength of gradient domination result. From my understanding, Theorem 1 is saying that gradient domination for every pseudoreward gives a global optimality result. This makes a lot of sense, as Hazan et al. [...]
>
> **Response to Question 1 (also to weaknesses 1-2):**
>
> To clarify: **Theorem 1 recovers the standard gradient domination result in linear RL**, as in Lemma 4.1 of Agarwal et al. (2021), when the pseudo-reward is constant and equals the true reward.
>
> - Importantly, our gradient domination inequality holds for each policy parameter $\theta$ with a distribution mismatch coefficient that depends on $\theta$, unlike in linear RL, where it is typically constant. This behavior also appears in prior work (e.g., Mei et al., ICML 2020 for softmax policies).
> - Crucially, the mismatch coefficient is **finite for any $\theta$**, provided the reference distribution $\mu$ has full support. So the result is not vacuous, for each $\theta$, the mismatch is of comparable order to the one in standard linear RL.
> - If one seeks a uniform upper bound over all $\theta$, the coefficient can be large, as you noted. However, this worst-case bound is still finite, and can be upper-bounded by the state space size if $\mu$ is uniform. It can also be upper bound by its maximum over the reward functions which are bounded by the range of pseudo-rewards using Assumption 4.
> - We also emphasize that this inequality is not used in the non-tabular setting (Sec. 4), where we instead rely on a different proof technique based on hidden convexity. Still, we believe this result is valuable to clarify the relationship between gradient dominance and RLGU more generally.
>
> > **Question 2:** Assumptions. Understanding the assumptions here is crucial to assess the generality ..... appear in Corollary 2? Shall not that be something like "for any precision ..."?
>
> **Response to Question 2 (also to weakness 3):**
>
> We appreciate the reviewer’s careful examination of our assumptions. While we build on several assumptions standard in RLGU and RL theory (e.g., smoothness, realizability, bounded pseudo-rewards), our contributions lie in how we leverage these to obtain new global guarantees, particularly in the large-scale setting. We will revise the paper to better explain the role of each assumption and when they are satisfied as suggested by the reviewer. We respond to the questions of the reviewer below.
>
> - **Realizability (Assumption 2 (ii))** is natural even for occupancy measures as we briefly argue in l. 227-231 (see appendix C for a proof), see response to Q6 below for more details. Regarding this assumption, we also refer the reviewer to Definition 1 and Lemma 16 of Huang et al. ICML 2023, Remark 3., Assumption D.1 in Huang and Jiang Neurips 2024 making the assumption. We can also replace our Assumption 2. (ii) with a low-rank MDP assumption.
>
> - **Assumption 4** is standard in the RLGU literature: see e.g. Hazan et al. 2019, Zhang et al. 2020-22, Barakat et al. 2023, Ying et al. 2023. This assumption captures most of the problems of interest in RLGU including pure exploration (using the smoothed entropy), learning from demonstrations (using the smoothed KL) as well as standard linear RL and CMDPs. Other entropic measures or $l^2$ (quadratic) losses are also possible. For instance the smoothed entropy defined as $H_{\sigma}(x) = - x log(x + \sigma)$ is $1/(2\sigma)$-smooth w.r.t. the infinity norm and has been used in RLGU, see e.g. Hazan et al. 2019, Lemma 4.3. We will add a discussion to the paper to mention these.
>
> - **Assumption 5:** As we mention it in l. 310-312, it was also used in  Zhang et al. (2021); Ying et al. (2023a); Barakat et al. (2023) and others. While this assumption holds for a tabular policy parametrization, it is delicate to relax it as we acknowledge and as it was mentioned in prior work. We note though that it is only a local assumption and this overparametrization assumption is natural when one looks at hidden convexity and the need to relate policies and occupancy measures. Trying to relax such an assumption is an interesting question.
>
> - **Regarding $\epsilon_{MLE}$**, note that this assumption is only made in theorem 3 to show how the bound depends on the estimation error of occupancy measures. In corollary 2, it is actually not required as this estimation error is driven to zero using Proposition 1. Indeed by picking the number of samples $n = \mathcal{O}(m/\epsilon^2)$ in corollary 2, the error $\epsilon_{MLE}$ is smaller than the desired function value gap accuracy $\epsilon$. Therefore there is no error floor that cannot be reduced in corollary 2. This is consistent with the fact that occupancy measures are realizable and we can approximate them arbitrarily well using enough samples (see Proposition 1). We will make this clearer in the paper.
>
>
> > **Question 3:** Occupancy measure estimation. Assuming i.i.d. sampling is somewhat problematic. It means n is actually the number of trajectory, not state-action pairs. The authors may consider making this more explicit in the paper, adding proper factors to the sample complexity results (at least an H factor shall be needed). Perhaps the results could look be refined by using more refined analysis for sampling and estimation from Markov chains. For the undiscounted case "Hoeffding's inequality for general Markov chains and its applications to statistical learning" Fan et al. 2021. For the discounted case "A tale of sampling and estimation in discounted reinforcement learning" Metelli et al. 2023.
>
> **Response to Question 3 (also to weakness 4):**
>
>  We do not require this iid assumption in the analysis, it is just to simplify the exposition in the main part and gain some space in the main part by deferring the sampling mechanism for states to appendix E.1. As for the horizon, we need to set it to $H = \mathcal{O}(1/(1-\gamma) \log(1/\epsilon))$. We thanl the reviewer for the comment, we will make this clearer.
>
> > **Question 4:**  I am not very familiar with the analysis of Barakat et al. 2023. [...] Aside from MSE and MLE step, what makes Barakat et al. analysis break for global optimality?
>
> **Response to Question 4 (also to weakness 1):**
>
> Apart from the use of smoothness, our global optimality analysis is completely different.
> - The first order stationarity analysis is based on exploiting smoothness of the objective alone as it is standard in smooth optimization.
> -  We need to exploit hidden convexity to obtain global optimality and we isolate and propagate errors induced by occupancy measure approximation in the PG method.
> - This leads to a function value gap recursion with errors satisfied by the optimality function value gap. See lines 814 to 831 for details compared to p. 41-44 in Barakat et al. 2023. In contrast, first-order stationarity immediately follows from telescoping the smoothness inequality.
> - In particular, our occupancy measure estimation and error control are different, our technical analysis is different and our guarantee is itself much stronger (our last-iterate global optimality vs best iterate first order stationarity).
>
> > **Question 5:** The results in the paper are provided for the discounted setting. In principle, can we use the same technical tools to study undiscounted settings, with finite or infinite horizon?
>
> **Response to Question 5.**  For the undiscounted setting with finite horizon, the answer is yes. One would need to be careful about modifying the occupancy measure estimators and the description becomes cumbersome if one decides to introduce non-homogeneous time-dependent state transitions as is customary in the finite horizon RL setting. For the undiscounted infinite horizon, if you think about the average reward setting, then extra care is needed as this case introduces some additional difficulties, note that PG convergence rates in this setting are very recent (see e.g. Kumar, Murthy et al. Global convergence of PG in average reward MDPs, ICLR 2025). This is an interesting direction for future work given the importance of the average reward.
>
> > **Question 6:** "Assumption 2 is satisfied for instance for the class of generalized linear models". Well, this is not supported when realizability is included in the assumption. Better to clarify in the text.
>
> **Response to Question 6 (Assumption 2 (ii)):** The realizability assumption holds in the case of low-rank MDPs. As we discuss in l. 230, state occupancy measures are linear in density features in low-rank MDPs as we prove it in appendix C. Beyond this low-rank case, we can also safely relax this realizability assumption in our analysis at the price of an error floor due to function approximation that cannot vanish if the true occupancy measures do not belong to our function approximation class. We will clarify this further in the text.
>
> > **Question 7:** The discussion of the related work looks mostly complete. Perhaps Cheung 2019 and the pair Prajapat et al. 2023, De Santi et al. 2024 could also be mentioned.
>
> **Response to Question 7: Additional related work.** Thank you for mentioning these references. For the last two, submodular RL differs from convex RL in nature. Nevertheless we will add them in our related work for completeness as they also propose an interesting approach to go beyond standard linear RL.
>
> We appreciate the reviewer’s careful and constructive feedback. We believe the revisions and clarifications above highlight the novelty and value of our contributions more clearly. We respectfully hope the reviewer will reconsider their evaluation in light of these updates.

---

> > ### Comment · Reviewer_hs2m · 2025-08-05
> >
> > Dear authors,
> >
> > Thank you for your detailed response. I am now more convinced on the value of the paper and raising my score to 5. I think the discussion of the assumptions, as well as the novelty of the analysis in technical terms, could be incorporated in the manuscript.

---

> > > ### Author Response · Authors · 2025-08-06
> > > **Thank you**
> > >
> > > Dear reviewer,
> > >
> > > Thank you for acknowledging our rebuttal. We greatly appreciate your efforts and valuable comments. We are glad to hear that our response helped clarify the value of our work. We will incorporate the discussion on assumptions and the technical novelty of our analysis into the manuscript.

---

### Official Review · Reviewer_sc7M · 2025-07-03

**Clarity:** 3
**Significance:** 3
**Originality:** 4
**Rating:** 4
**Confidence:** 4

**Summary:**

This paper studies reinforcement learning with general utilities (RLGU), where in a discounted Markov decision process (MDP), the objective is to maximize a utility function $F$ defined on the occupancy $\lambda^\pi$. Specifically, $\lambda^\pi (s, a)$ is the normalized discounted occupancy of $(s, a)$ under policy $\pi$, and the utility is $F(\lambda^\pi)$. This work first establishes the gradient domination theorem for tabular MDPs and $F$ which is concave in $\lambda$ (not $\theta$, the parameter of $\pi$). This theorem can further imply $\epsilon^{-1}$ convergence for policy gradient (PG) methods. Then, when the occupancy falls inside a $m$-dimensional parametric class, the authors propose a two-stage PG algorithm, which iteratively does MLE for occupancy and policy update using the estimation. Under mild assumptions, the authors establish a $m \epsilon^{-2}$ convergence to stationary points for general utility, and a $m \epsilon^{-4}$ sub-optimality gap for concave utility.

**Questions:**

1. Can you discuss about the cases of unknown $F$? For example, if the MDP is tabular and we only know that $F$ is concave, how much difficulty will removing the knowledge of $F$ impose on deriving a similar guarantee?

2. Can you provide real-world examples aside from standard RL that RLGU plays an important role? Is there any existing environments or benchmarks for this problem?

**Ethical Concerns:**

["NO or VERY MINOR ethics concerns only"]

**Final Justification:**

The rebuttal addresses my concerns of the applications of general utility RL. However, the difficulty of unknown $F$ is still evasive, as the authors state that this is common in all previous works. Given this, I'll maintain my rating of 4.

**Limitations:**

One limitation is the applicability of RLGU, which is from the problem side. This type of problem seems to mainly attract theoretical interests and lack practical significance, as I'm not aware of existing environments or benchmarks modeling RLGU. The other limitation is the assumption of known utility, which is from the algorithm side. There lacks a discussion on the necessity of this assumption.

**Quality:**

3

**Strengths And Weaknesses:**

### Strengths
- **Theories:** Establishes gradient domination properties for RLGU objectives, connecting them to recent advances in standard RL theory. First work to provide global optimality guarantees for RLGU beyond tabular settings. Results cover both concave and general utilities. Theorem 3 is a last-iterate global convergence guarantee which previous works fail to provide.
- **Algorithm design:** PG-OMA is easily scalable to function classes beyond tabular.


### Weaknesses
- **Assumption on known utility:** This paper assumes that $F$ is known to calculate $\nabla_\lambda F(\lambda)$, which is used in the "policy gradient" in Equations (4) and (5). While this assumption is without loss of generality for standard RL (known reward does not make problem easier), it might significantly reduce problem difficulty for more general (or even concave) utilities.
- **Lacking empirical verification:** Did not provide empirical validations of the proposed algorithms, on either synthetic or practical environments. This weakens the importance of RLGU to real-world problems.

---

> ### Author Rebuttal · Authors · 2025-07-31
>
> We thank the reviewer for their time in evaluating our work and for their useful feedback.
>
> >**Weakness 1:** Assumption on known utility: This paper assumes that  is known to calculate , which is used in the "policy gradient" in Equations (4) and (5). While this assumption is without loss of generality for standard RL (known reward does not make problem easier), it might significantly reduce problem difficulty for more general (or even concave) utilities.
>
>
> **Response to Weakness 1:** As with most works in RLGU, our setting assumes access to the gradient of the known utility function $F$, which reflects the problem’s objective and is typically task-defined (e.g., entropy, divergence or quadratic, see also Mutny et al., 2023 for other examples). This assumption is standard in the literature (see e.g. Hazan et al. 2019, Zhang et al. 2020-2021, Bai et al. 2022, Barakat et al. 2023, Ying et al. 2023a,b), and allows us to isolate the optimization challenges inherent to non-linear, global objectives in RL. We will expand this discussion in the revised version.
>
> > **Weakness 2:** Lacking empirical verification: Did not provide empirical validations of the proposed algorithms, on either synthetic or practical environments. This weakens the importance of RLGU to real-world problems.
>
> **Response to Weakness 2:**
>
> We agree that practical validation of our algorithm in function-approximation settings is an important direction, and we are actively exploring this. Our current focus is to first establish a theoretical foundation — particularly global optimality guarantees — which had not previously been achieved for RLGU.
>
> > **Question 1:** Can you discuss about the cases of unknown F? For example, if the MDP is tabular and we only know that F is concave, how much difficulty will removing the knowledge of F impose on deriving a similar guarantee?
>
> **Response to Question 1:** We need some form of feedback on the function $F$ to be optimized. If it is fully unknown and we do not have access to any form of feedback, then there is not much we can do. In our case, we only have access to a gradient oracle (gradient can be computed) as it is standard in the literature. If the reviewer is thinking about weaker forms of feedback such as zeroth order access to $F$ or some bandit feedback, then we believe this offers some interesting opportunities for future work using zeroth order optimization techniques or regret analysis for bandit feedback. We leave such investigations for future work.
>
> > **Question 2:** Can you provide real-world examples aside from standard RL that RLGU plays an important role? Is there any existing habitats or benchmarks for this problem?
>
> **Response to Question 2:**
>
> - As we briefly discuss in the introduction, there are plenty of applications for RLGU beyond standard RL. We provide more concrete applications in what follows building on the examples in the introduction:
>
>     - (1) *Pure exploration* of state spaces (e.g. Hazan et al. ICML 2019), concrete applications include autonomous exploration of robots such as drones or Mars rovers where agents are tasked with mapping unknown environments without specific downstream tasks, in this reward-free setting, agents seek to maximize information gain, coverage, or novelty; another compelling application in this setting is scientific discovery: Robots or agents exploring chemical or material design spaces to uncover new compounds, protein discovery as well;
>    - (2) *Imitation learning* (e.g. Ho and Ermon, NeurIPS 2016), applications in this setting include learning from diverse human demonstrations and social navigation where robots learn to imitate various crowd navigation strategies,
>    - (3) *Experiment design* (e.g. Mutny et al., AISTATS 2023) in science where an agent chooses experiments to maximize expected informativeness,
>    - (4) *Diverse skills discovery* (e.g. Eysenbach et al. ICLR 2019) in robotic tasks with the unsupervised emergence of diverse skills, such as walking and jumping.
>   - (5) *Pluralistic alignment* (e.g. of LLMs), see e.g. Chakraborty et al. ICML 2024, 'MaxMin-RLHF: Alignment with Diverse Human Preferences'.
>
> - Regarding benchmarks, for instance in (Ho and Ermon, 2016) and (Eysenbach et al. 2019), they adapt the standard continuous control task benchmark of Brockman et al., 2016 in standard RL. Existing works test their approach on specific applications. We are not aware of an established benchmark for RLGU with diverse objectives. We believe that such a benchmark would be clearly valuable for advancing the impact of RLGU in practice. Our contributions are primarily theoretical though.
>
> We appreciate the reviewer’s careful and constructive feedback. We believe the revisions and clarifications above highlight the novelty and value of our contributions more clearly. We respectfully hope the reviewer will reconsider their evaluation in light of these updates.

---

> > ### Comment · Reviewer_sc7M · 2025-08-07
> >
> > The rebuttal addresses my concerns of the applications of general utility RL. However, the difficulty of unknown  is still evasive, as the authors state that this is common in all previous works. Given this, I'll maintain my rating of 4.

---

> > > ### Author Response · Authors · 2025-08-08
> > > **Response**
> > >
> > > We thank the reviewer for acknowledging our rebuttal and for the opportunity to provide further clarification regarding the knowledge of $F$.
> > >
> > > We would like to clarify that in most applications of RLGU known in the literature, the function $F$ is available, and its gradient can therefore be computed directly. Thus, in the majority of scenarios, access to $F$ (and its gradient in particular) is not a restrictive requirement. As discussed in the introduction, we elaborate below on several important examples of RLGU, focusing specifically on the knowledge of $F$:
> > >
> > > - Pure exploration (see e.g. Hazan et al. 2019): $F$ is the negative entropy function.
> > >
> > > - Imitation learning and apprenticeship learning (see e.g. Ho and Ermon 2016): $F$ is the Kullback-Liebler divergence between the state-action occupancy measure induced by a policy and expert demonstrations, or a differentiable distance between the aforementioned distributions.
> > >
> > > - Constrained RL (see e.g. Zhang et al. 2024): $F$ is the sum of a linear function (as in standard RL) and a penalty function such as the smoothed log barrier function.
> > >
> > > - Experiment design (see e.g. Mutny et al. 2023): see A and D-design objectives which are differentiable.
> > >
> > > More generally, the user can choose their own function $F$ to define their RLGU problem of interest.
> > >
> > > While our focus is on the known-$F$ setting, which encompasses several important RLGU applications,  the unknown-$F$ regime (e.g., under zeroth-order feedback, where zeroth-order optimization techniques may apply) is an interesting direction for future exploration, as highlighted in our previous response.

---

### Official Review · Reviewer_px47 · 2025-07-05

**Clarity:** 3
**Significance:** 4
**Originality:** 4
**Rating:** 4
**Confidence:** 5

**Summary:**

Reinforcement learning with general utilities is an important topic in the RL community. This paper establishes global optimality guarantees of PG methods for RLGU in which the objective is a general concave utility function of the state-action occupancy measure. In the tabular setting, this paper provides global optimality results using a new proof technique building on recent theoretical developments on the convergence of PG methods for standard RL using gradient domination.

**Questions:**

Is the assumption of bounded gradients truly necessary, or could the results be generalized under gradient moment assumptions (e.g., bounded variance or sub-Gaussian tails?

Given that global optimality is approached as the inner-loop step size shrinks, how does this affect the sample efficiency and convergence speed in practice? Does the theory support any practical heuristics for tuning inner step sizes?

**Ethical Concerns:**

["NO or VERY MINOR ethics concerns only"]

**Final Justification:**

I have no further questions.

**Limitations:**

See weakness.

**Quality:**

4

**Strengths And Weaknesses:**

Strengths:

This paper establishes tight bounds on the optimality gap using clear mathematical derivations and well-stated assumptions.

While prior works have analyzed convergence or approximation properties of RLGU in practice or for specific architectures, this work is among the first to offer a general analysis of global optimality, which is an important supplement in the RLGU community.

Weaknesses:

The analysis considers a single-step gradient update in the inner loop. While the authors briefly mention multi-step generalization, no formal analysis is provided. This limits applicability to practical implementations using multiple inner-loop steps.

Experiments are very limited. While they serve to confirm the theory, they offer little practical insight into real-world performance or limitations.

---

> ### Author Rebuttal · Authors · 2025-07-30
>
> We thank the reviewer for their time and feedback. We are glad the reviewer appreciates our contributions and finds them an ‘important supplement in the RLGU community’.
>
>
> > **Weakness 1:** The analysis considers only a single‑step gradient update in the inner loop. Although multi‑step updates are briefly mentioned, no formal analysis is provided, limiting applicability to implementations that use multiple inner‑loop steps.
>
> **Response to Weakness 1:** We do not mention multi-step generalization, but if you are alluding to the occupancy measure step or the PG step as inner-loop, Our analysis can easily be adapted to handle multiple gradient steps, this can only improve our results. Once can also use multiple gradient ascent steps in MLE in practice.
>
> > **Question 1:** Is the assumption of bounded gradients truly necessary, or could the results be generalized under weaker moment assumptions (e.g., bounded variance or sub‑Gaussian tails)?
>
> **Response to Question 1:** We do not make a boundedness assumption on the gradients of the objective function **with respect to its policy parameters**. The only boundedness assumption we make is that of the pseudo-reward functions in Assumption 4, as $\nabla_{\lambda} F(\lambda)$ plays the role of a pseudo-reward in our RLGU setting and it is a natural reward boundedness assumption that is standard in both RL as well as in RLGU, see e.g. Hazan et al., 2019; Zhang et al., 2020, 2021; Barakat et al., 2023; Ying et al., 2023a. We actually control the variance of our PG estimator precisely in p. 21-22 eqs. 31-35 using a minibatch of trajectories and a number of samples for occupancy estimation.
>
>
> > **Question 2:** Since global optimality is approached as the inner‑loop step size shrinks, how does this affect sample efficiency and convergence speed in practice? Does the theory support any practical heuristics for tuning the inner step size?
>
> **Response to Question 2:** In practice, we can choose constant steps to ensure a good neural network function approximation of occupancy measures in MLE by running multiple steps of SGD with tuned constant stepsizes and then using the estimates to perform PG ascent with a constant step and appropriate minibatch size.  We provide an analysis of the total sample complexity in corollaries 1 and 2 with the appropriate step sizes needed to obtain in-expectation first order stationarity for non-concave utilities and in expectation global optimality for concave utilities. Decreasing the step size as prescribed in theory is needed even in practice to reduce the training error.
>
> We appreciate the reviewer’s feedback. We hope our clarifications address the concerns raised, and respectfully ask the reviewer to reconsider their evaluation in light of these updates.

---

> > ### Comment · Reviewer_px47 · 2025-08-06
> > **Official Comment by Reviewer px47**
> >
> > Thank you for the detailed responses. The additional explanations and experiments resolve some of my concerns and questions.

---

### Note · Authors · 2025-08-15

We sincerely thank all reviewers for their time in assessing our work and for their valuable feedback, which we will use to further improve the manuscript. The reviews were consistently positive and acknowledged that our rebuttal addressed the main concerns. We will incorporate the suggested related work and expand the discussion on assumptions and technical novelty.

Beyond standard RL, the RLGU framework unifies a range of important applications, including pure exploration, imitation learning, experiment design, diverse skills discovery, among others. Our work establishes global optimality guarantees for policy gradient methods in RLGU with concave utilities, in both tabular and non-tabular settings. In the tabular case, we prove a novel gradient domination result for RLGU objectives—a structural property that connects them to recent advances in policy gradient theory for standard RL. For large state-action spaces, we propose a PG algorithm that uses occupancy measure approximation via function approximation. This is the first work to provide global optimality guarantees for RLGU beyond tabular settings, with iteration and sample complexity that scale with the dimension of the function approximation parameters rather than the size of the state–action space.

Promising future directions include extensions to more general policy parameterizations, continuous state–action spaces, and the average-reward setting, which is attracting growing interest in theoretical RL. We hope our work will foster further research in RLGU and its diverse applications across theory and practice.

---

### Decision · Program_Chairs · 2025-09-17

**Decision:**

Accept (poster)

**Comment:**

This paper analyzes the policy gradient method for reinforcement learning with general utilities and establishes global optimality guarantees. Specifically, it proves global optimality for policy gradient when the utility is concave and the MDP is tabular, using the gradient domination technique. The paper then extends the analysis beyond the tabular setting by estimating the state-action occupancy measure via maximum likelihood within an approximating class.

To the best of my knowledge, this is the first work to provide global optimality guarantees for reinforcement learning with general utilities (RLGU) beyond the tabular case, and it establishes a tight bound. Most reviewers agreed that the paper merits acceptance, and after carefully reviewing the rebuttal and discussion, I share this view. That said, I recommend that the authors incorporate the reviewers’ feedback into the final version of the paper, with particular attention to:

1- Including real-world examples that highlight the practical significance of the presented analysis.

2- Adding discussion on the limitations of the analysis, especially for the continuous case, and clarifying the assumptions made.